# Determinants of HIV Testing during Pregnancy among Pregnant Sudanese Women: A Cross-Sectional Study

**DOI:** 10.3390/bs12050150

**Published:** 2022-05-17

**Authors:** Ibrahim Elsiddig Elsheikh, Rik Crutzen, Ishag Adam, Salah Ibrahim Abdelraheem, Hubertus W. Van den Borne

**Affiliations:** 1Department of Health Promotion, Care and Public Health Research Institute (CAPHRI), Maastricht University, 6200 MD Maastricht, The Netherlands; rik.crutzen@maastrichtuniversity.nl (R.C.); b.vdborne@maastrichtuniversity.nl (H.W.V.d.B.); 2Sudanese Public Health Association (SPHA), Khartoum 11111, Sudan; 3Department of Obstetrics and Gynecology, Unaizah College of Medicine and Medical Sciences, Qassim University, Unaizah 56219, Saudi Arabia; ishagadam@hotmail.com; 4Department of Obstetrics and Gynecology, Omdurman Maternity Hospital, Khartoum 24983, Sudan; dr.s.ibrahim2015@gmail.com

**Keywords:** PMTCT, pregnant women, health promotion

## Abstract

More than 90% of children who are HIV positive were infected via mother-to-child transmission (MTCT). In Sudan, HIV testing rates during pregnancy remain low. This study aimed to understand the key determinants of HIV testing and their association with pregnant women’s intention to undergo HIV test during pregnancy. A cross-sectional survey was conducted among 770 Sudanese pregnant women attending Antenatal care (ANC) visits at maternity hospitals. Based on the flow of antenatal care attendants, the calculated sample size was proportionally allocated to the hospitals. Doctors were most influential regarding pregnant women’s decision to undergo an HIV test during pregnancy (78.8%). Younger women were more likely to be tested. Most participants (68.9%) had high susceptibility with respect to HIV. Nearly half (48.3%) had a positive attitude towards HIV testing. Self-efficacy with regard to HIV testing was high (59.1%). Women with high self-efficacy and perceived susceptibility were more likely to have a greater intention to be tested for HIV. No significant association was found with perceived severity and stigma. Our study shows that the intention to undergo HIV testing among pregnant women is influenced by doctors and associated with self-efficacy and perceived susceptibility, which are important avenues for future intervention efforts.

## 1. Introduction

A total of 38.0 million people are living with HIV and AIDS worldwide, of which 1.7 million people became newly infected with HIV in 2019 [1]. More than 17.0 million are women, and 1.8 million are children less than 15 years old [2]. More than 90% of children who are HIV positive were infected via mother-to-child transmission (MTCT) [2,3]. MTCT can occur during pregnancy, at delivery or during breastfeeding, but the respective frequencies in these times are difficult to be determined [4]; some findings indicated that more than half of the transmission probably occurred late in pregnancy or during labor and delivery [5]. Sub-Saharan Africa is the region most affected globally, with the highest rates of new HIV infections [3]. AIDS is considered the number one cause of death for adolescents in Africa [6]. Of the 1.8 million children under age 15 years living with HIV and AIDS, only half are receiving treatment. In addition, only half of babies at risk for HIV are tested by the recommended age of two months [6]. Global efforts are being made to achieve the strategic goal of eliminating MTCT and thereby ensuring zero new HIV infections among children [7]. Prevention of mother-to-child transmission (PMTCT) interventions has proven to effectively reduce MTCT [8]. These interventions usually include HIV testing, enrollment for treatment during early stages of pregnancy, labor management, information on breastfeeding, and pediatric care [9,10,11]. 

The HIV epidemic in Sudan is driven mainly by heterosexual transmission. It has been estimated that among all new HIV infection cases, 59% occur in women of reproductive age [9]. This indicates the importance of scaling up PMTCT services in the country [9]. PMTCT services began in Sudan in 2005 and have increased steadily. In 2012, Provider Initiated Testing and Counseling (PITC) was introduced as well [12,13]. In a review article on the maternal HIV infection in Sudan, despite 89% Antenatal care (ANC) access in urban areas, HIV testing during pregnancy is <1% [9]. Moreover, the level of PMTCT service uptake is deficient even though PMTCT services including HIV test are free of charge. This can be attributed mainly to the inefficient implementation of PITC because of health system factors such as the reluctance of health care providers to offer HIV testing to pregnant women and lack of accountability and weak integration of HIV testing as a part of routine check-ups for pregnant women [12]. In addition, incomplete implementation of the opt-out strategy regarding HIV testing, limited training of health care providers, and delays in shifting from vertical to integrated programs are also known as key factors affecting the scale-up of PMTCT services [9,14]. However, minimal research about PMTCT has been carried out in Sudan that has focused on identifying the determinants of HIV voluntary counseling and testing (VCT) services among pregnant women [13,15]. 

A study was conducted in Khartoum Hospital, located at the center of Khartoum city and one of the country’s oldest and most prominent hospitals. It has a section for perinatal medicine with senior specialists that offer services almost free of charge; 55.9% (562/1005) of the pregnant women interviewed had some information about MTCT, but only 30.3% (305/1005) had been tested [15]. In a similar study in Ethiopia, 57.5% of pregnant women attending ANCs had full knowledge about MTCT of HIV, but only 17.4% knew about PMTCT of HIV/AIDS [16]. A study in Kenya revealed that lower educational level among women was associated with less acceptance of HIV testing during pregnancy [17]. The reasons for the refusal of HIV testing among pregnant women in Kenya mainly were related to HIV/AIDS stigma. Women were afraid that they would lose their partners if they tested positive for HIV [18]. This was also the case in other African countries such as Tanzania [19] and South Africa [20]. In terms of disclosure, in a qualitative study in Sudan, the majority of women stated that an HIV-positive person should not let others know his/her status, because s/he will get stigmatized (e.g., s/he could lose his/her job and be forced to leave the family home) [13]. Another study in Kenya indicated that pregnant women did not get tested because of stigma or fear of disclosure (64.2%), followed by reasons related to denial or not wanting to know their HIV status (56.8%), and the husband’s disapproval (15.1%) [21]. Shame and blame or being judged by others were significantly associated with decreased HIV testing during pregnancy in the same study in Kenya. In Uganda, women in urban areas with some education and at younger ages showed higher acceptance of VCT [22]. In a study in Ethiopia, subjective norms perceived that behavioral control, attitude, and perceived susceptibility were associated with the high intention to undergo HIV testing during pregnancy; however, perceived severity did not show any associations [23]. Perceived high personal susceptibility to HIV/AIDS, confidentiality, partner involvement, and self-efficacy were all associated with intention to accept VCT in Northern Tanzania [24]. 

There are some previous studies that identified factors associated with HIV testing, but not within a Sudanese context. This study aimed to understand the key determinants of pregnant women’s intention to undergo HIV test during pregnancy. The findings of this study are used to develop future health promotion interventions aimed at increasing the HIV testing rates during pregnancy in Sudan. In this study, we derived the examined determinants from the reasoned action approach, the extended parallel process model, and the socio-psychological view of stigma. Both psychosocial and contextual factors were taken into consideration. Given the overlap between the reasoned action approach and the extended parallel process model, self-efficacy and perceived behavior control were considered the same in this problem-driven research. A distinction was made between how these factors concerned the mother and how they concerned the child. 

## 2. Materials and Methods

A cross-sectional survey was conducted among women attending ANC visits at five major public maternity hospitals in greater Khartoum in 2014. The selection of the hospitals was made purposively because these were the only sites that provided complete PMTCT services at that time. A total of 770 pregnant women were selected at these five PMTCT sites. The sampling frame was the total number of pregnant women who visited the five hospitals for maternity follow-up as per ANC registers. The total number of pregnant women per hospital was identified from ANC records. Then, based on the flow of antenatal care attendants, the calculated sample size was proportionally allocated to the hospitals. Distributed by hospital and study region (representative for their size): as follows; Omdurman Maternity Hospital (580 women), Al-Saudi Maternity Hospital (82), Turkish Maternity Hospital in Khartoum South (50), Khartoum Hospital Maternity Department (44), and Bahri Hospital (14). Pregnant women were interviewed (aiming for N = 770, based on a margin of error of 0.05 and 10% nonresponse) [25] by trained data collectors upon entering the ANC clinic at each maternity hospital for ten consecutive days. Every pregnant woman who arrived at the clinic during these times had the chance to be recruited unless she opted out. The process continued until the target sample size was reached. Of 1114 women invited to participate in the study, 790 (71%) accepted. Only 14 pregnant women opted out in the middle of the interview, and the questionnaire was discontinued; these women were replaced by newly recruited respondents from the reserve group, as per the research protocol. 

### 2.1. Recruitment of Participants and Procedure

Recruitment eligibility was every pregnant woman attending an ANC clinic at the five hospitals unless she opted out. HIV status was not part of the selection criteria. At the time of the interviews, 96.4% of women indicated that they had been pregnant at least once. Verbal informed consent was obtained from all women before the interview started. The women were told that there was no need for names and that the information collected was purely for research purposes. It was emphasized that participation was entirely voluntary. Interviews were conducted in Arabic, the native language of respondents. 

The questionnaire, which was in Arabic, was pre-tested among 50 pregnant women (10 from each hospital) and was found to be understandable. No major changes were made based on the pretest. The questionnaire included five major sections: sociodemographic information, determinants, services for testing and counseling, sources of information, and care and treatment. The factors that were explored in this study were derived from multiple socio-cognitive theories: the extended parallel process model [26] (perceived severity, perceived susceptibility, and response efficacy), the reasoned action approach (attitude, subjective norms, perceived behavioral control), and the sociopsychological view of stigma (personal responsibility and norm-violating behavior) [27]. Given the overlap between the reasoned action approach and the extended parallel process model, self-efficacy and perceived behavior control were considered in this problem-driven research. The constructs measured were:

Attitude: This construct was measured with five items measured on a 5-point Likert scale, ranging from ‘strongly agree’ to ‘strongly disagree’. Four items were positive statements with regards to HIV-testing and one was negative. The negative statement was recoded for interpretation purposes. An example statement was: ‘HIV-testing is helpful to make informed decisions about breastfeeding’. Assessment of the internal structure of the scale showed an omega of 0.73 (95% CI: 0.75–0.80). The score for attitude was the mean of the five item scores.

Subjective norm: Two items were used to measure the subjective norm. For both statements a 5-point Likert scale was used ranging from ‘strongly agree’ to ‘strongly disagree’. One statement regarding the subjective norm for example was: ‘My community/relatives would blame me if I tested positive for HIV.’ Reliability analysis showed a Spearman’s Brown coefficient of 0.73. Furthermore, the mean score for subjective norm was calculated.

Perceived Severity: This variable was measured through the following two statements: ‘HIV is severe for me as a pregnant woman’ and ‘HIV is severe for my child.’ Both statements were measured on a 5-point Likert scale, ranging from ‘strongly agree’ to ‘strongly disagree’. Reliability analysis showed a Spearman’s Brown coefficient of 0.84. Furthermore, the mean score for perceived severity was calculated.

Perceived Susceptibility: This variable was measured through the following two statements: ‘I am likely to be affected by HIV/AIDS’ and ‘My child is likely to get affected by HIV/AIDS.’ The answers of the scale, measured on a 5-point Likert scale, ranged from strongly agree to strongly disagree. Furthermore, the scale was deemed reliable with a Spearman’s Brown coefficient of 0.86. Moreover, the mean score for perceived susceptibility was calculated.

Self-Efficacy: Given the overlap between the reasoned action approach and the extended parallel process model, self-efficacy and perceived control were overlapping and self-efficacy was measured trough the following statement: ‘I am able to take an HIV-test while I’m pregnant’. The statement was measured on a 5-point Likert scale ranging from ‘strongly agree’ to ‘strongly disagree’. In addition to the above, we have also assessed the influence of others to take HIV test or not and the perceived advantages and disadvantages. 

HIV-related stigma: This variable was measured through the statement: ‘Experiencing stigma would make it difficult for me to take an HIV test’. This statement was measured on a 5-point Likert scale and ranged from ‘strongly agree’ to ‘strongly disagree’.

The full questionnaire is available at https://osf.io/46wep/?view_only=06a58b2c77bf4ad49948ee4d4a9a4553 (accessed on 16 April 2022). It is also attached as a supporting document.

### 2.2. Data Analysis

The dependent variable, intention to take an HIV test during pregnancy, was measured by the following item: “Do you intend to have an HIV test in the future (including current pregnancy)”. The independent variables include sociodemographic and cognitive variables. Sociodemographic variables were age, residence, religion, education, occupation, income, marital status, and woman’s current living arrangement. Cognitive variables were attitude, subjective norms, self-efficacy, perceived severity, perceived susceptibility, and experienced stigma. 

We started by exploring the sociodemographic and cognitive variables by means of descriptive statistics. Subsequently, bivariate and multivariate logistic regression analysis were conducted using IBM SPSS version 23 (IBM Corp., Armonk, NY, USA). This was done in two steps. The first step consisted of a simple logistic regression model with the sociodemographic and cognitive variables as predictors for the dependent variable (i.e., intention to take an HIV test during pregnancy). The second step consisted of a multiple logistic regression model with the same variables as predictors for these dependent variables. Odd ratios (ORs) and 95% confidence intervals (CI) were calculated. *p* < 0.05 was considered statistically significant. 

## 3. Results

### 3.1. Subsection

#### Sociodemographic Characteristics

A total of 770 Sudanese pregnant women were included in the statistical analysis. More than half of the participants were between ages 20 and 29 years. The mean (SD) age was 28.1 (6.2) years. Eighty-five percent resided in and around Omdurman. Omdurman is one of the three cities that comprise Khartoum the capital of Sudan. It is the most populated city in the country, and thus also in Khartoum. Omdurman lies on the west bank of the River Nile, opposite and northwest of the Khartoum city. Furthermore, 32.7% had a low education level, which entails that a participant did not go to school or only finished primary school. Thirty-six percent of women had an intermediate level education (secondary and/or occupational school) and 31% had a higher education level (university and/or postgraduate). More than 80% of participants were housewives and nearly all were Muslim and currently married. See Table 1. 

### 3.2. Knowledge about MTCT

More than half of the pregnant women (58.3%) had insufficient knowledge about MTCT. 

#### 3.2.1. Perceived Advantages and Disadvantages of HIV Testing during Pregnancy 

Most pregnant women understood the advantages of receiving HIV testing during pregnancy, 73.8% believed that HIV testing during pregnancy has no disadvantages. However, some of them (23.5%) still associated testing with an element of fear, suspicion, stigmatization, and most of them did not actually receive the test.

#### 3.2.2. Influence on the Decision to Receive or Refuse an HIV Test 

Doctors had the most influence on pregnant women’s decision to undergo an HIV test during pregnancy (78.8%). The husband had some influence (19.4%), followed by the woman’s mother (1.3%); midwives had no influence (1.1%). More than half of the pregnant women (58.6%) indicated that no one would disapprove of their decision to have an HIV test if they wished; however, 20.2% indicated that their husbands would disapprove. See Table 2 below. 

#### 3.2.3. Experience with HIV Testing and Intention to Undergo HIV Testing in the Future 

Most of the pregnant women (66.8%) indicated that they had never been tested for HIV; only 31.0% stated that they had been previously tested. Among women who were offered the test, 91.5% said the test was offered to them by doctors and 6.5% said they asked for the test themselves. When asked if they intended to undergo HIV testing in the future (including current pregnancy), 66.6% of the women indicated their intention to be tested. According to logistic regression using bivariate analysis, there was an association with age and intention to undergo HIV testing. Younger women are more likely to accept HIV testing (OR 1.043, 95% CI 1.014–1.073). No significant associations were found with education, occupation, religion, monthly income, and HIV testing intention. Upon multivariate analysis, the age of the pregnant woman and with whom the women are currently living were found to have significant associations. Moreover, no significant associations were found with residence, education, occupation, religion, and HIV testing intention. Refer to Table 3 below. 

### 3.3. Cognitive Factors 

Most participants (69.6%) had high perceived severity regarding HIV. Most participants (68.9%) also had high susceptibility with respect to HIV. Nearly half of respondents (48.3%) had a positive attitude towards HIV testing. Self-efficacy with regard to HIV testing was high among most participants (59.1%). Additionally, 66.6% of the women reported that they intended to be tested for HIV. 

The following significant associations were found in the bivariate analysis. Women with higher self-efficacy (OR 1.856; 95% Cl 1.582–2.177, *p* < 0.001) and higher perceived susceptibility (OR 1.417; 95% Cl 1.212–1.656, *p* < 0.001) were more likely to have a higher intention to be tested for HIV. Significant associations were found in subjective norms, women were more likely to not get themselves tested (OR 1.161, 95% CI (1.020–1.322), *p* = 0.023) when others engage in this behavior. However, no significant association was found with attitude, experienced stigma and perceived severity. Refer to Table 4. 

## 4. Discussion

Our study revealed that 58.3% of the pregnant women had insufficient knowledge about MTCT. This was in keeping with research in Ethiopia in which only 17.4% of participants had knowledge about PMTCT of HIV/AIDS [16]. In this study, doctors were the most influential regarding the decision to receive an HIV test during pregnancy. This is in line with previous findings in qualitative study in Sudan [13]. This implies the importance of training and sensitizing doctors to improve the HIV testing rates during pregnancy. Most of the pregnant women perceived advantages in undergoing HIV testing during pregnancy, 73.8% saw no disadvantages and showed an intention to perform the test; however, most of them did not actually receive the test. This is in line with findings from a previous study in Sudan in which, among 72.8% of pregnant women who indicated their willingness to take an HIV test, only 30.3% actually did so [15]. 

Our study applied determinants derived from multiple theories to explain HIV testing intention among pregnant women in Sudan. The results show that some socio-cognitive determinants had a significant relation with HIV testing intention, namely, self-efficacy, perceived susceptibility and subjective norms. This is in line with similar findings from studies in Ethiopia [23] and Tanzania [24]. However, attitude, perceived severity and stigma did not show any significant associations with HIV testing intention. From a programmatic point of view, these results imply the importance of focusing on both personal determinants and community-based interventions to increase HIV testing during pregnancy. HIV stigma did not show a significant relation with the intention to undergo HIV testing. This contradicts many other studies in which stigma was viewed as a key barrier to receiving an HIV test [18,19,20]. In the current research, the constructs of HIV stigma and subjective norms showed some overlap; therefore, it is important that future studies attempt to make a clear distinction between the two constructs.

Our findings showed that age and current living arrangement were associated with the intention of pregnant women to undergo HIV testing; younger women were more likely to be tested, which is similar to findings from Uganda [22]. This might be because younger women had a higher chance of access to information and knowledge about MTCT, better knowledge regarding the advantages of HIV testing, as well as ability to make good decisions to go for HIV testing, which increases their motive to be tested. Another factor may be that older women had been previously tested for HIV. For the variable current living arrangement, women who lived with their extended family had a greater intention to receive HIV testing compared with women who lived in a nuclear family. This might be attributed to the influence on women by members of an extended family, such as mothers and mothers-in-law. 

One of the limitations of this study is the focus on the intention to undergo HIV testing and not on the behavior itself of being tested. Therefore, it is essential to obtain data on testing behavior in future studies to obtain a more accurate view of the current situation in Sudan. Despite the high number of women who said they intend to be tested for HIV in the current study, the actual uptake of HIV testing is still deficient in Sudan; intention is not being translated into real behavior. In the literature, this phenomenon has been referred to as the intention–behavior gap [28]. There may be numerous reasons for Sudanese women not to be tested for HIV, even though they have an intention to do so. In addition to factors at the individual level, there could be environmental factors, such as access to health facilities and financial limitations [29]. To bridge this gap, it is important to examine testing behavior itself and what results in intentions translating into actual HIV testing behavior.

## 5. Conclusions

Our study concluded that the intention to undergo HIV testing among pregnant women in Sudan is influenced mainly by recommendations from health care providers, particularly Doctors. If Doctors offer the test, pregnant women will do it. The results show that some socio-cognitive determinants had a significant relation with HIV testing intention, namely, self-efficacy, perceived susceptibility and subjective norms. We also found that attitudes, perceived severity and HIV stigma did not have associations with HIV testing intention. It is recommended that future programs aiming at increasing PMTCT uptake in Sudan focus on these determinants.

## Figures and Tables

**Table 1 behavsci-12-00150-t001:** Characteristics of the study population.

Variables	N (%)
Age 20–29 years	415 (53.9)
30–39 years	268 (34.8)
40+ years	32 (4.2)
Highest level education	
Low (can read and write, finished primary school)	252 (32.7)
Intermediate (completed secondary or occupation school)	279 (36.2)
High (has university degree or post graduate)	239 (31.0)
Occupation	
Housewife	625 (81.2)
Working	145 (18.8)
Residence	
Omdurman	661 (85.5)
Khartoum	14 (1.8)
Bahri	95 (12.3)
Religion	
Muslim	762 (99.0)
Christian	8 (1.0)

**Table 2 behavsci-12-00150-t002:** Perceived Advantages and dis-advantages of taking HIV test during pregnancy.

What Do You See as the Advantages of You Taking an HIV Test during Pregnancy?	N (%)	What Do You See as the Dis-Advantages of You Taking an HIV Test during Pregnancy?	N (%)
Know my status	331 (41.5%)	Has no disadvantages	588 (73.8%)
Safeguarding the child	297 (37.3%)	Fear and Uncertainty	102 (12.8%)
Build trust	233 (29.2%)	Create suspicion	66 (8.3%)
Feel control	116 (14.6%)	Stigmatization	19 (2.4%)
Has no advantages	24 (3%)		

**Table 3 behavsci-12-00150-t003:** Socio-Demographic predictors for HIV test among pregnant women using bivariate and multivariate analyses.

	Bivariate Analysis			Multivariate Analysis		
Variable	OR	95% CI	*p*	OR	95% CI	*p*
Location	0.983	0.853–1.133	0.812	0.963	0.827 –1.122	0.632
Age	1.043	1.014–1.073	0.004	1.017	1.015–1.079	0.003
Religion	0.846	0.163–4.399	0.842	0.796	0.150–4.227	0.789
Education	1.073	0.951–1.211	0.253	1.032	0.898–1.186	0.661
Occupation	1.091	0.947–1.257	0.226	1.126	0.876–1.201	0.752
Ever been pregnant	2.961	0.184–47.583	0.444	0.537	0.033–8.868	0.664
Who do you currently live with	1.727	1.119–2665	0.014	1.905	1.209–3.001	0.005

**Table 4 behavsci-12-00150-t004:** Socio-cognitive predictors for HIV test among pregnant women using bivariate and multivariate analyses.

	Bivariate Analysis			Multivariate Analysis		
Variable	OR	95% CI	*p*	OR	95% CI	*p*
Attitude	0.937	0.586–1.499	0.786	1.007	0.499–2.029	0.985
Subjective norms	0.861	0.756–0.980	0.023	0.726	0.527–1.000	0.050
Self-efficacy	1.856	1.582–2.177	<0.001	2.071	1.488–2.883	<0.001
Perceived severity	0.968	0.845–1.109	0.643	0.813	0.561–1.180	0.276
Perceived susceptibility	1.417	1.212–1.656	<0.001	1.178	0.811–1.712	0.390
Experienced stigma	0.959	0.850–1.082	0.499	1.093	0.824–1.499	0.538

## Data Availability

Data supporting reported results can be requested.

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
