# Peer review of "Determinants of HIV Testing during Pregnancy among Pregnant Sudanese Women: A Cross-Sectional Study"

_behavsci, 2022, doi:10.3390/bs12050150_

Round 1
Reviewer 1 Report
The chosen topic and contents used for this article are interesting and regard a significant survey launched among Sudanese pregnant women to understand the key determinants of their intention to undergo HIV test.
Since the manuscript has been considerably improved, it is currently adequate and stimulating, with a clear and sound presentation of methods, results and conclusions.
Author Response
Point 1: The chosen topic and contents used for this article are interesting and regard a significant survey launched among Sudanese pregnant women to understand the key determinants of their intention to undergo HIV test.
Since the manuscript has been considerably improved, it is currently adequate and stimulating, with a clear and sound presentation of methods, results and conclusions.
Response 1: Thank you very much for your encouraging words. We are delighted to read your review.
Reviewer 2 Report
I appreciate the authors’ response to the reviewers and revision of the manuscript.
Now I have a question to the authors:
- I am afraid that ending the conclusion section with a topic of stigma is inappropriate, because the manuscript itself is not focused on it and includes a result of no statistically significant association among stigma and a willingness to HIV test, although of course I know the authors would like to talk about it. Please reconsider this.
Author Response
Point 1: I am afraid that ending the conclusion section with a topic of stigma is inappropriate, because the manuscript itself is not focused on it and includes a result of no statistically significant association among stigma and a willingness to HIV test, although of course I know the authors would like to talk about it. Please reconsider this.
Response 1: Please provide your response for Point 1. (in red). Thank you, we have amended our conclusion based on the reviewer feedback. It is now reading as follows:
Our study concluded that the intention to undergo HIV testing among pregnant women in Sudan is influenced mainly by recommendations from health care providers, particularly Doctors. If Doctors offer the test, pregnant women will do it. The results showed that some socio-cognitive determinants had a significant relation with HIV testing intention, namely, self-efficacy, perceived susceptibility and subjective norms. We also found that attitudes, perceived severity and HIV stigma did not have associations with HIV testing intention. It is recommended that future programs aiming at increasing PMTCT uptake in Sudan focus on these determinants

This manuscript is a resubmission of an earlier submission. The following is a list of the peer review reports and author responses from that submission.
Round 1
Reviewer 1 Report
The chosen topic and contents used for this article are interesting and regard a significant survey launched among Sudanese pregnant women to understand the key determinants of their intention to undergo HIV test. The aim is reasonably clear, although the authors did not sufficiently highlight why and how they designated the determinants from multiple socio-cognitive theories, namely the extended parallel process model, the reasoned-action approach, and the socio-psychological view of stigma.
In the paragraph Materials and Methods, the authors did not sufficiently describe the questionnaire structure, the chosen scales for the responses and how many items they used to measure each construct. Moreover, the full questionnaire itself is not easily accessible at the provided link: https://osf.io/tms5q/. The text is not very fluid, since the different concepts are not well defined and linked together. Furthermore, a more rigorous conceptualization is mandatory, so that readers can easily follow the logic and appreciate the contents. In addition, the paragraph Conclusions should be enriched by more details.
In summary, the study could be potentially adequate and stimulating, but lacks a clear and sound presentation - in particular of methods, results and conclusions - in order to highlight the real contribution to the field.
Reviewer 2 Report
Please find an attached file.
